# The Multifaceted Role and Regulation of Nlrp3 Inflammasome in Colitis-Associated Colo-Rectal Cancer: A Systematic Review

**DOI:** 10.3390/ijms24043472

**Published:** 2023-02-09

**Authors:** Roxana Zaharie, Dan Valean, Calin Popa, Alin Fetti, Claudiu Zdrehus, Aida Puia, Lia Usatiuc, Diana Schlanger, Florin Zaharie

**Affiliations:** 1Department of Gastroenterology, University of Medicine and Pharmacy “Iuliu Hatieganu”, 400347 Cluj-Napoca, Romania; 2Regional Institute of Gastroenterology and Hepatology “O. Fodor”, 400162 Cluj-Napoca, Romania; 3Department of General Surgery, University of Medicine and Pharmacy “Iuliu Hațieganu”, 400347 Cluj-Napoca, Romania; 4Department of Anesthesiology, University of Medicine and Pharmacy “Iuliu Hațieganu”, 400347 Cluj-Napoca, Romania; 5Department of Family Medicine, University of Medicine and Pharmacy “Iuliu Hatieganu”, 400347 Cluj-Napoca, Romania; 6Department of Pathophysiology, University of Medicine and Pharmacy “Iuliu Hațieganu”, 400347 Cluj-Napoca, Romania

**Keywords:** NLRP3 inflammasome, colitis associated colo-rectal cancer, prognosis, activation and inactivation pathways

## Abstract

Colitis-associated colo-rectal cancer remains the leading cause of mortality in inflammatory bowel diseases, with inflammation remaining one of the bridging points between the two pathologies. The NLRP3 inflammasome complex plays an important role in innate immunity; however, its misregulation can be responsible for the apparition of various pathologies such as ulcerative colitis. Our review focuses on the potential pathways of upregulation or downregulation of the NLRP3 complex, in addition to evaluating its role in the current clinical setting. Eighteen studies highlighted the potential pathways of NLRP3 complex regulation as well as its role in the metastatic process in colo-rectal cancer, with promising results. Further research is, however, needed in order to validate the results in a clinical setting.

## 1. Introduction

Colitis-associated colo-rectal cancer (CA-CRC) remains one of the main causes of mortality in inflammatory bowel diseases (IBD), being responsible for up to 15% of IBD-associated deaths [1]. As yearly worldwide IBD incidence rates are increasing, with a high degree of variability and presenting a higher incidence in developed regions, the rates of CA-CRC are also likely to increase [1,2]. One of the primary risk factors of CA-CRC is considered to be the duration of IBD, which can present with an earlier age of onset, in comparison with non-IBD-related colo-rectal cancer (CRC) [2]. The cumulative risk of CA-CRC is estimated to be around 2% at 10 years of disease onset, up to 20% at 30 years of disease activity [3].

One of the common grounds between IBD and colo-rectal cancer is represented by inflammation, mainly a disregulation of the inflammation pathways, which is present in all phases of the carcinogenesis [2]. Although the exact mechanism that links chronic inflammation to the development of CRC, especially CA-CRC is not fully established, most of the recent studies focus on either an aberrant or defective response of the innate immune system, which is considered to be the primary mechanism in chronic inflammation of the intestinal mucosa [3,4].

As a first line of host defense, the innate immune system presents itself with an array of mechanisms that are activated via the pattern-recognition receptors (PRR) by various agents. The most crucial components of the innate immune system are inflammasomes, a series of multiprotein oligomers responsible for the activation of various inflammatory response pathways. The primary roles of inflammasomes consist of: (a) removal of the pathogenic agent, (b) maintaining tissue homeostasis, and (c) removal of tumoral cells via an adaptive immune response [5,6,7,8].

NLRP3 is a protein encoded by the NLRP3 gene, which is located on the long arm of the first chromosome that functions as a PRR which is responsible for the detection of pathogen-associated molecular patterns (PAMPs) [5]. Mutations of the NLRP3 inflammasome complex play an important role in the development of dominantly inherited autoinflammatory diseases, and its deregulation has been linked to carcinogenesis [9,10]. Recent studies have focused on the various pathways that can either downregulate or inactivate NLRP3 in order to cement its role in various pathologies [11,12,13,14].

This review will summarize the role of the NLRP3 inflammasome complex in the development of CA-CRC and summarize studies that report on the downregulation of NLRP3 activity, in order to develop therapeutic strategies in IBD as well as in CA-CRC. This can further ramp up the role of NLRP3 as a prognostic factor in the aforementioned pathologies as well as identify various methods of treatment in the chronic inflammation of the intestinal tract. Our focus will not be on the mechanisms of action, as this was discussed in the previous reviews [11,15,16].

## 2. Methods

A systematic review was designed according to the Preferred Reporting Items for a Systematic Review and Meta-analysis of Individual Participant Data Criteria (PRISMA). This review intended to gather all of the recent evidence regarding the role of NLRP3 in the pathogenesis of CA-CRC, as well as the pathways of activation or inactivation.

Original research studies, as well as experimental studies and clinical trials, were included that had the NLRP3 inflammasome as the primary focus of the study. All retrospective, prospective, comparative, cohort studies, and clinical trials were included in the review. Commentary articles, review articles, editorials, and articles written in different languages than English were excluded. The reason for the exclusion of the commentary articles and editorials is the lack of a detailed methodology and also the presence of subjective opinions. Although review articles were not included, a search of the reference list in the reviews was performed in order to search for other articles that might be suitable for our review.

A systematic search was performed on three of the major databases: PubMed, Scopus, and Embase, using the following terms for searching “NLRP3 colitis cancer" and “NLRP3 colo-rectal cancer”. No automation tools were used for the search, and a thorough analysis of the articles was performed by three different reviewers. The reviewing selection process focused on the articles over the last 10 years (August 2012–August 2022). The selection process was divided into two stages: the first stage focused on reading the titles and abstracts to verify the inclusion or exclusion criteria. The selected articles were moved on to the second stage, and by analyzing the contents of the articles they were selected for further reviewing. For the second stage, all authors of this review were consulted to minimize the bias or any discrepancies. All articles that were selected after this two-stage process were included into our review. Data collected were summarized into tables, which highlight the year of the article, the primary authors, the type of the study, the design of the study, and the main outcomes of the study, as well as the potential implications and limitations of the study.

### 2.1. Search Strategy

A total number of 248 articles were identified (70 articles in the PubMed database, 84 articles in the Scopus database, and 88 articles in the Embase database). Out of these articles, 150 articles were identified as duplicates and therefore excluded. Furthermore, we excluded 56 articles based on the title and the abstract. The remaining 42 articles were submitted for a full read and review, out of which 27 were selected based on the proposed criteria. This process is detailed below using the PRISMA flowchart, represented in Figure 1.

### 2.2. Included Studies

The included studies are listed in Appendix A. The selected studies were published between 2014 and 2022, with most of the studies being published in the last 3 years (21/27 studies—74%). The majority of the studies were experimental studies focusing on three major aspects of the review: means of upregulation of NLRP3 complex, means of downregulation of NRLP3 complex, and metastatic activity in CRC and CA-CRC of the NLRP3 complex. Out of all the studies, 19 of them also had cell cultures (19/27—70.3%) with a primary focus on human colon carcinoma cells. In most of the experimental studies, mice were used with induced UC and CRC via dextran sulfate sodium (DSS) [14].

## 3. Results and Discussion

### 3.1. General Considerations

It is known since 2002 that inflammasomes play a key component in innate immunity as they are responsible for releasing proinflammatory cytokines (notably IL-1β and IL-18) via activation of the Caspase-1 protease [15]. Inflammasomes are divided into three major families (NLR, PYHIN, and ALR), with NLRP3 being a key member of the first family (NOD-like-receptor family, pyrin domain-containing protein 3). NLRP3 is triggered by the presence of PAMPs (pathogen-associated molecular patterns) as well as substances that are present due to metabolic imbalances, such as uric crystals [16]. These substances can trigger the inflammasome complexes, which will in turn trigger the assembly of the cytokines. Inadequate activation of the NLRP3 inflammasome complexes can be responsible for a series of inflammatory or metabolic-related diseases, from gout to Alzheimer’s disease; therefore, knowing the potential pathways of activation and inhibition of the complex can provide further methods of treatment [17,18,19]. It is also known that misregulation of NLRP3 is associated with the development of UC as well as CA-CRC [10,20].

Although the triggers for activating the pathways of the NLRP3 inflammasome are still under debate, with some of the key elements being suggested in the previous studies, there are two known types of activation [10,17,20]. The canonical activation involves the presence of a PAMP, which generates an upregulation of the NLRP3 transcriptional mRNA. In addition, the presence of DAMPs (damage-associated molecular patterns) trigger the involvement of the apoptosis-associated speck-like protein (ASP), a key component of the inflammasome complex, which in turn will recruit and activate the Caspase-1 through its caspase activation and recruitment domain component (CARD). Finally, caspase-1 acts as a catalyst in the generation of the biologically active mature IL-1β and IL-18 by cleaving the IL-1 family precursor, which in turn generates the inflammatory response [21].

The non-canonical pathway is activated by intracellular DAMPs, with this pathway, however, being caspase-11 dependent (ROS) [17,20,22]. Upon activation, its path merges with the canonical pathway. One of the main objectives of this review is to highlight the potential regulation pathways of the NLRP3 inflammasome complex, for further treatment methods.

Major pathways of activation and inhibition of the NLRP3 inflammasome complex that are involved in the development of CA-CRC are highlighted in Figure 2.

### 3.2. Activation of the NLRP3 Complex

Activation of the NLRP3 inflammasome complex has been controversial. Although early studies have shown a potential protective effect in the evolution of ulcerative colitis [23,24], recent manuscripts have shown an entirely different approach, with the majority of the studies leaning towards its proinflammatory effect, with the apparition of CA-CRC being considered as the end-stage of the inflammation mechanisms [13,14].

One of the effects it plays on the development of colorectal cancer is its association with the epithelial–mesenchymal transition (EMT), which plays an important role in the progression of the disease [25,26]. Histopathological findings as well as immunohistochemistry in patients with CRC have identified elevated levels of NLRP3, ASC, procaspase 1, and IL-1β compared with the control tissues, which indicate an up-regulation of the complex in the presence of colo-rectal cancer [25,26]. In addition, its activation was in direct relation to the stage of the disease, and the relative mRNA expression was higher in the higher tumoral grades [25,27]. NLRP3 is directly correlated with EMT markers such as vinectin, N-cadherin, and MMP9, and inversely correlated with E-cadherin. These results support the statement that NLRP3 is involved in the development of CRC [25,26,27].

In addition, a number of studies have focused on NLRP3 as a prognostic marker [28,29]. Analysis of frozen human tissue samples and mouse xenograph models using the human colon carcinoma cell lines HCT116 and RKO demonstrated that NLRP3 was upregulated in human colon carcinoma tissue [29]. Histochemical analysis of 60 pairs of human colon adenocarcinoma and paracancerous microarray samples showed NLRP3 upregulation in 49 adenocarcinoma samples and downregulation in 11 adenocarcinoma samples [29]. Notably, other inflammasomes, such as NLRC4, can also be associated with inflammation-induced tumorigenesis [30]. In addition, an analysis made by Wang et al. highlighted a link between the upregulation of NLRP3 with the mTOR-S6K1 pathway in CRC tissues, thus suggesting a potential therapy target in mTORC1-targeted resistant patients [31].

A high-cholesterol diet can promote azoxymethane (AOM)-induced colorectal cancer in mice through activation of the NLRP3 inflammasome, as cholesterol crystals are interpreted as danger signals responsible for priming the inflammasomes. In addition, cholesterol crystals enhanced the ASC speck formation and molecular interactions between ASC and NLRP3, supporting the role of NLRP3 inflammasome activation in cholesterol-induced effects. In addition, the occurrence of cholesterol crystals bound by phagocytosis is the prerequisite for activation of a cytosolic NLRP3 inflammasome. Thus, crystal uptake and lysosomal rupture are required for NLRP3 inflammasome activation induced by cholesterol crystals. This supports the fact that a high-cholesterol diet can induce inflammation in the colon through NLRP3 inflammasome activation. Deletion of NLRP3 abolishes the accumulation of cholesterol crystals in macrophages which in turn decreases the expression of IL-1B. In addition, a high-cholesterol diet can lead to constitutive activation of the Beta-catenin pathway, which can contribute to the initiation and progression of sporadic CRC- or IBD-associated carcinogenesis. Beta-catenin levels were increased in mice with high-cholesterol diet. Proliferating cell nuclear antigen, target genes for Beta-catenin, was elevated in high-cholesterol-diet treated mice, but not elevated in the abolished NLRP3 mice, therefore suggesting the fact that cholesterol activates beta-catenin signaling, which is required for the NLRP3 activation [32].

High stromal levels of transmembrane protein 176B, a cationic channel, is associated with low overall survival in patients with CRC. Since TMEM176B inhibits NLRP3 activation, this stands in seeming contrast to the results of the studies presented above. However, TMEM176B appears to primarily affect immune infiltrating cells and not the primary tumor [33]. Segovia et al. highlighted the key role of the TMEM176B channel in anti-tumoral therapy; thus, an upregulation of the molecule leads to inhibition of the NLRP3/caspase-1/IL-1 pathway, which is relevant for dendritic cell activation and cytotoxic response [34]. This is relevant especially in the tumor microenvironment because an inhibition at this level may lead to suppression or inhibition of the metastatic potential in cancer cells [35].

In the context of perineural invasion in CRC and CA-CRC, one of the neurotransmitters frequently involved in the tumoral microenvironment after perineural invasion is 5-hydroxytryptamine (5-HT), also known as serotonin [36]. At the gastrointestinal level, 5-HT is produced by enterochromaffin cells, with its purpose being the regulation of gastrointestinal motility and secretions. 5-HT is produced by tumoral cells via their autocrine function in an attempt to regulate the microenvironment [37]. It has been shown that colorectal cancer cells present an upregulation of 5-HT as well as tryptophan hydroxylase 1 (TPH1), which is a biosynthesis rate-limiting enzyme [36,37,38]. Gut microbiota contribute to the modulation of serotonin signaling, with this statement being supported by (a) changes in the composition of bacteria post-antibiotic treatment; (b) changes in serotonin levels correlated to specific bacteria; and (c) decreased serotonin transporter expression may be associated with a shift in gut microbiota from homeostasis to inflammatory-type microbiota [36,37].

In addition, 5-HT can enhance the production of IL-1β in macrophages on certain colorectal cancer cell lines. In addition, by testing whether the blocking of the 5-HT receptors will downregulate the production of IL-1β, only the 5-HTRA3 receptor significantly impaired the production [38]. Therefore, the HTRA3 receptor is primarily responsible in the development of proinflammatory cytokines in CRC and CA-CRC. This complex, in turn, enhances NLRP3 inflammasome activation [37]. Blocking HTRA3 receptors with TPS or shRNA significantly impaired the inflammasome complex. Through the calcium/calmodulin-dependent protein kinase II (Ca^2+^/CaMKIIα complex), 5-HT/HTRA3 can regulate NLRP3 phosphorylation. Finally, we can conclude that a 5-HT-NLRP3-positive loop can facilitate tumoral growth, with 5-HT having an important role in the GI tract inflammation as well as CRC development [37,38,39]. This loop can be furthermore targeted as a potential line of treatment against the development of CA-CRC.

*Akkermansia Municiphila* is a species of human intestinal mucine-degrading bacterium who plays an important role in colorectal cancer pathogenesis. Studies have shown a decrease in A. municiphila in patients with CRC, and supplementing with it suppressed colonic tumorigenesis in APC-induced mice [40,41]. In addition, *A. municiphila* facilitated enrichment of M1-like macrophages in an NLRP3-dependent manner, both in vitro and in vivo. Thus, an NLRP3 deficiency in macrophages can attenuate the tumor suppressive effect of *A. municiphila*, as the results were compared between NLRP3-deficient mice and wild type mice; this is in conjecture with the aforementioned statements that activation of NLRP3 in immune cells is beneficial [5,6,7,8]. Therefore, in NRLP3-deficient mice, the tumor-protective effect of A. municiphila is abolished [40].

Bacterial cells can also produce misregulation of the NLRP3 complex, especially *E. coli* [42,43]. Intracellular *E. coli* is more prevalent in patients with inflammatory bowel diseases, especially UC. In addition, intracellular *E. coli*, isolated from ileal cells, can present a high resistance to macrophage activity [42,43]. These bacteria can induce interleukin production which can stimulate proinflammatory activity via the macrophages. Since *E. coli* is more prevalent in IBD patients compared with control subjects, especially the strains that present with macrophage resistance, this may be associated with proinflammatory production. This might represent a challenge since only *E. coli* strains associated to IBDs are associated with the virulence genes and survival inside the macrophages; therefore, the proinflammatory status is much more difficult to tackle [43,44]. However, some polymorphisms of the NLRP3 gene which are associated with a lower expression of the protein can lead to dysfunctional activity of the inflammasome during the early disease activity which can lead to a weak inflammatory response. This weak inflammatory response in turn will lead to sustained inflammation due to the impairment of bacterial clearance in the intestinal mucosa [43,44,45,46]. 

Noteworthy bacterium that activates the NLRP3 inflammasome, thus having an influence on the development of colorectal carcinoma, is *Porphyromonas gingivalis* (*P*. *gingivalis*), a periodontal pathogen, currently associated with various digestive cancers [47,48,49]. Using FISH and IHC staining, significant quantities of *P. gingivalis* were found in the fecal matter of patients with colorectal cancer [48]. It can also serve as an independent predictor of colorectal cancer aggression, with elevated levels being positively associated with poor survival, and shorter recurrence-free survival. In addition, using an experimental model on mice that can develop intestinal tumors can promote colorectal tumorigenesis and also produce a proinflammatory microenvironment [47,49].

In contrast to the studies presented above, Li et al. (2021) report that activation of the NLRP3 inflammasome complex via induction of dextran/sulfate sodium can have a protective effect by presenting more remission symptoms and less loss of body weight, with tumor burdens significantly lost in the IOP group [50]. Therefore, IOP can reduce CA-CRC severity in an experimental model. When NLRP3, IL-1 β, and IL-18 levels were evaluated, an increased expression in mice treated with IOP was noticed compared with the control group. In addition, the antitumoral effect of IOP is mediated by the NLRP3 inflammasome [51]. Other studies have also reported the anti-tumorigenic functions of NLRP3 [52]. Thus, the role of NLRP3 in CA-CRC is not fully understood and NLRP3-mediated anti-and pro-tumorigenic mechanisms remain a critical area of research [52].

### 3.3. Inhibition of the NLRP3 Complex

The primary focus for the inhibition of the NLRP3 inflammasome complex rests on finding an adequate method of treatment or control of the progression of IBDs. Over the last decade, the number of articles that focus on this aspect have increased exponentially, highlighting the need for inhibitors with the potential to become novel therapies.

Intestinal microbiota play a primordial role in the development of IBD as well as in the development of CA-CRC. Some bacteria, as highlighted earlier, can enhance the inflammatory phase as well as exhibit significant drug resistance during chemotherapy or anti-cancer treatment. In addition, disorders of the intestinal microbiota are associated with abnormal or impaired immune activity [53,54].

It is known that commensal bacteria-mediated NRLP3 inflammasome activation can induce IL-1B secretion [55]; however, *E. faecalis* can attenuate this response [56]. The levels of caspase-1 and mature Il-1B in infections with *E. coli* or *P. mirablis* were decreased in cells pre-treated with *E. faecalis* compared with untreated cells. This suppressive effect appeared to be dose-dependent and the effect against commensal bacteria was observed in THP-1 cells infected with different doses of the previously mentioned agents. Therefore, *E. faecalis* may have an attenuative effect on fecal content or commensal bacteria-induced NLRP3 activation. Bacterial phagocytosis is also required for bacteria-induced NLRP3 inflammation; however, *E. faecalis* interferes with phagocytosis, thus attenuating the activation of the NLRP3 inflammasome through a different mechanism [56].

Furthermore, *E. faecalis* significantly reduced DSS-induced weight loss and diarrhea in treated mice, compared with the wild mice, but not compared with the NLRP3-defficient mice. The cleaved caspase-1 levels and mature IL-1B were significantly reduced in *E. faecalis*-treated mice than in the control group, thus showing that *E. faecalis* treatment can protect mice from DSS-induced colitis through an NLRP3-dependent manner. Since inflammation is a major risk factor for CRC development, *E. faecalis* treatment significantly inhibited the number of DSS-induced colon tumors per mouse. Therefore, it may also exert protective effects against CRC, without acting on pre-existing CRC [55,56].

*Bacteroides Fragilis* has been shown to restrict CA-CRC by negatively regulating the NLRP3 inflammasome axis [57]. Bacteroides are symbiotic bacteria in the human gastrointestinal tract; however, they can exhibit worse responses when in other parts of the body. In UC, the levels of *B. fragilis* are significantly lower than in the control group, which can be associated with the increasing levels of inflammation in the prevalent disease. In addition, by reproducing dysbiosis in mice (via broad-spectrum antibiotic administration), significant weight loss and poor nutritional status have been observed, along with changes in the histological activity, with the presence of inflammation, hyperplasia, and dysplasia in some cases [57,58]. Therefore, dysbiosis of the gastrointestinal tract can be associated with the enhancement of CA-CRC [58]. Furthermore, administering *B. fragilis* in mice with dysbiosis highlighted a significant improvement compared with the control group, as the specimens’ nutritional status was greatly improved. In addition, their feces showed greatly increased contents of short-chain fatty acids, namely butyrate. Thus, *B. fragilis* has been shown to promote the secretion of butyrate, which can restrict the development of CA-CRC via the downregulation of NLRP3 [58,59], although its role has been contested by some studies, supporting the claim that the excess of butyrate can inhibit the proliferation of stem cells [60,61].

In another study, Atractylenolide I inhibited NLRP3 inflammasome formation and thus the IL-1 β secretion in AOM/DSS (azoxymethane/dextrane sodium sulfate) mice and reduced the viability of cell cultures (HCT116 and SW480) by inducing apoptosis [62]. It significantly reduced the expression of intestinal NLRP3, caspase-1, and ASC. It also inhibited the DRP1-mediated mitochondrial fission, which in turn can inhibit NLRP3 in bone-marrow-derived macrophages. Upregulation of DRP-1 can reverse the effect of the inflammasome activation that can lead to mitochondrial fission. In cases of immunoprecipitated proteins with anti-ASC, authors managed to greatly inhibit the expression by using Atractylenolide I; however, in cases of DRP1 overexpression, it had no effect on the expression of the NLRP3 inflammasome [63]. Therefore, Atractyleonide I can suppress NLRP3 inflammasome expression as well as reverse the mitochondrial damage caused by their activation; however, this activity is dependent on the expression of DRP-1 [62].

Small molecules, such as GL-V9 (5-hydroxy-8-methoxy-2-phenyl-7-(4-(pyrrolidin-1-yl)butoxy) 4 H-chromen-4-one) can be involved in the inhibition or downregulation of NRLP3 inflammasome through autophagy, and can lower the effects of DSS-induced colitis via an AMP-activated protein kinase (AMPK/FOXO3 gateway) by upregulating TRX-1, while also triggering autophagy through this activation [64].

Caffeic acid phenyl ester (CAPE) is a bioactive extract usually found in various dietary supplements, fruits, and grains, which is known for its antioxidant properties. Although some early studies focused on alleviating the CA-CRC in DSS-induced mice by suppressing the mieloperoxidase activity, the main mechanism through which CAPE inhibits CAC is unknown [65]. However, some authors managed to inhibit NLRP3 protein expression in vitro and in vivo using CAPE in addition to adding a protective effect against CA-CRC in AOM/DSS-induced mice [66]. Although CAPE does not affect the mRNA levels of NLRP3, it decreases its protein levels facilitating ubiquitination, therefore suggesting that CAPE manifests inhibitory effects suppressing ROS production in THP-1 cells. Through its antioxidant effects, CAPE can recuperate ROS, and can also suppress its production at the transcriptional level (via NF-kB activation) [65,66]. The main issue with CAPE remains the dosage, due to it having a low level of acute oral toxicity; however, further studies are required to assess the efficiency or safety before clinical use [66,67].

Another natural extract, an alkalide dipertenoid named Andrographolide, has been reported to have a protective effect in AZM/DDS-induced colon cancer in mice through inhibiting the NLRP3 inflammasome via mitophagy mediation, therefore decreasing the IL-1B levels. The consensus of NLRP3 regulation is that the inflammasome is positively driven by reactive oxygen species and negatively regulated by mitophagy. In this case, however, andrographolide-driven mitophagy leads to reduction in the damaged mitochondria, which inactivates the NLRP3 inflammasome complex, thus ameliorating CA-CRC in mice. In addition, andrographolide can induce autophagy in various cells through different ways. Therefore, it has a dual protective effect against CA-CRC [68].

Another flavonoid commonly used for its immunomodulatory effects, quercetin, can have anticancer potential through its modulation of cellular functions such as apoptosis or inhibition of cell proliferation. A study focused on fermenting quercetin with *Lactobacilius* strain LY219, which enhanced the cytotoxicity of 5FU in resistin-treated CRC cells. Resistin is a cytokine known for its role in obesity-related inflammation and cancer proliferation. CRC patients present elevated levels of resistin. Fermented quercetin inhibits NLRP3 expression, which can promote the loss of cell viability, apoptosis, and autophagy. Therefore, quercetin can be used as a compound therapy with 5-FU due to it being reported to increase the efficacy of anticancer drugs and inhibit the gene expression of multi-drug resistance 1 [69].

Arctigenin, another natural extract has proven to play a significant role in the inhibition of CA-CRC by downregulating the NLRP3 activation as well as the fatty acid metabolism (FAO) in AOM/DSS-induced CAC in mice [70]. In this study, all of the mice that were treated with arctigenin survived. It is established that this compound significantly reduced the pathological changes, and reduced the number of adenocarcinomas inside the mucosa, decreasing the number of abnormal cells. In addition, arctigenin downregulates fatty acid oxidation, but not glycolysis, during the assembly of NLRP3 inflammasomes. CPT1, the rate-limiting enzyme in FAO has been shown to inhibit the inflammasome assembly as well. However, the activity is not disrupted by the Arctigenin treatment. Despite this, an overexpression of the CPT1 can put the NLRP3 assembly into overdrive, resulting in ASC oligomerization and enhanced caspase-1 activation which ends with the assembly of IL-1 β. Arctigenin can also disrupt the NRLP3 assembly via inhibiting the alpha-tubulin acetylation since it mediates the spatial arrangement of the mitochondria and can cause insufficient assembly of the ASC within the NLRP3 complex [70,71].

MicroRNAs have proven to play a key role in inhibition of the NLRP3 inflammasome complexes, with one example being miR-22. Its effects have been studied on human colon cancer cells by exposing them to a miR-22 mimic, miR-22 inhibitor, and control mimics, respectively [72]. MiR-22 overexpression in human cancer cells, especially in HT116 cells, can suppress its cellular activity, whereas its inhibition would promote cell activity. In addition, cell migration and invasion were notably higher in the miR-inhibitor group and significantly lower in the mimic group. In HT116 cells, miR-22 targets the NLRP3 inflammasome, inhibiting cell proliferation, migration, and invasion, and furthermore controlling epithelial–mesenchymal transformation [72,73]. The main limitations of this study were the use of a single strand of cancer cells (HCT116) and the analysis of the levels of IL-1 β using ELISA assays; however, the effects of the stimulation of cells with IL-1β were not compared [72].

A recently developed small molecule, RRx-001, with the purpose of NLRP3 inhibition demonstrated enhanced antitumor effects when used in combination with regorafenib. Regorafenib is a salvage option with short-term OS and PFS benefits, with an array of side effects that yield poor toleration of the treatment. In combination with regorafenib, the agent is much more effective than agent alone, which is mainly due to the attenuation of regorafenib toxicity in vivo; however, further clinical trials are required to highlight this benefit [73,74].

### 3.4. The Impact of NLRP3 in Liver Metastases in CRC and CA-CRC

Even though the link between inflammation and cancer has been proven in the last two decades, there are few articles in which the role of inflammation is established in a metastatic background. Although approximately 1/3 of the patients with CRC develop metastases, more than 75% of them are located in the liver [75]. Even though aberrations at the cellular level are initiators in cancer, the same aberrations can either develop or limit tumoral growth. Both IL-1 β and IL-18 exhibit multiple effects in inflammation as well as in tumorigenesis. However, they can also exert an anti-tumoral immunity besides their tumoral promotion effect; these effects are specific to various tissues and can only appear under certain conditions [76].

The discussion between pro- and anti-metastatic growth still remains controversial. Some authors suggest that inflammasome complex stimulation through inflammation and 6rly cancer will trigger NK cell maturation, therefore developing a tumoricidal activity [77]. IL-18 tumoricidal activity was linked to both cytotoxic T cells and NK cells in previous studies. NK cells are important in the control of liver metastatic growth. The anti-metastatic function, however, requires FasL-resistant (Fas-ligands) tumoral cells. Thus, the control of hepatic metastasis will occur regardless of the strength of the adaptive immune system [77]. IL-18 will act on NK cells in order to control the development of liver metastases. This activity is independent of the IFN-gamma activity in mediating cell suppression of metastasis that presents with FasL-resistant cells [77,78].

However, more authors support the fact that NLRP3 inflammasome with its role in inflammation and tumorigenesis will eventually drive the metastases towards the liver. Due to the overexpression of NLRP3, macrophages will drive up the production of proinflammatory interleukines, which through the macrophage activity will promote cell migration [79,80]. Through a bone-marrow-derived macrophage study (BMDM) in human cancer cells (SW480), a significant increase in cell migration was noted. These effects can be diminished by an IL-1 β-neutralized antibody. Moreover, IL-1 β can directly promote cell migration and increase expression of EMT markers. Blocking NLRP3 signaling is shown to suppress cell migration. Therefore, macrophages can influence the CRC migration through activation of the NLRP3 inflammasomes, increasing IL-1 β secretion which can ulteriorly be decreased by blocking NLRP3 signaling [23,79]. In addition, a study has shown that the expressions of the macrophage surface marker CD-68 and the proinflammatory cytokine IL-1B are positively correlated. Therefore, NLRP3 inflammasome activation in MΦs-CRC crosstalk can promote invasion, migration, as well as EMT of the colo-rectal cancer cells in vitro; thus, its overexpression is positively associated with advanced AJCC stage and a poor prognosis [80].

Through an experimental model of CRC metastases, cancer-induced NLRP3 activation can lead to NK cell tumoricidal activity. Mice with deficiency in components of the NLRP3 inflammasome presented a higher metastatic burden compared with the control group. Cancer-derived factors that activate the NLRP3 response will lead to IL-18 maturation that will increase the production of IFNγ, which in turn promotes a Th1 response, stimulating the cytotoxic activity of NK cells. IL-18 induces tumoricidal activity independently of adaptive immunity effectors. By downregulating the NLRP3 signaling, the production of IFNγ is not altered; however, the FasL surface expression is decreased. Therefore, the NLRP3 inflammasome in Kupffer cells boosts tumoricidal activity through FasL-induced apoptosis [81].

### 3.5. Study Limitations

All the included studies provide novel information regarding the upregulation and downregulation of the NLRP3 inflammasome complex, which are important to highlight. However, there are some drawbacks within the main topic and article selections. All articles have an experimental design; therefore, the results need to be validated and assessed by further studies in order to be replicated in a potential clinical setting. In addition, the articles which present cell culture studies can present themselves in different settings rather actual clinical trials.

Having these limitations taken into consideration, however, it is important to acknowledge the results obtained by the current studies, which can encourage further research into the topic.

## 4. Conclusions

NLRP3 inflammasome plays a crucial role in the inflammatory mechanisms in IBD, as well as in tumorigenesis and tumoral development and cell migration. Despite this, the discussion regarding the regulation of the inflammasome complex is still on the table, which may suggest that an underexpression or an overexpression may exert a different effect during an inflammatory bowel disease. Therefore, it is crucial to understand the complexity and variety of activation and inhibition pathways, even at an experimental level. A variety of factors, from gut microbiota, diet, or plant extracts can either upregulate or downregulate the inflammasome complex, with some factors being dependent on another co-factor. There is still a debate whether NLRP3 has pro-metastatic or anti-metastatic activity, as well as a tumorigenic or anti-tumorigenic activity; therefore, more studies are required in this regard. An important role of the inflammasome complex as a prognostic biomarker should be underlined, especially in the development of CA-CRC, which could further develop the prognosis of colo-rectal cancer. In addition, most of the recent studies focus on the development of targeted treatments, albeit experimental, which could prove beneficial, especially in individualized therapies. This opens up a new avenue for treatments, not only in the realm of colo-rectal cancer, but also in the realm of inflammatory bowel diseases and inflammation-related diseases as well. Notwithstanding the number of recent articles, there is still room for improvement and understanding of the innate immune barriers in the development of colitis-associated cancer.

## Figures and Tables

**Figure 1 ijms-24-03472-f001:**
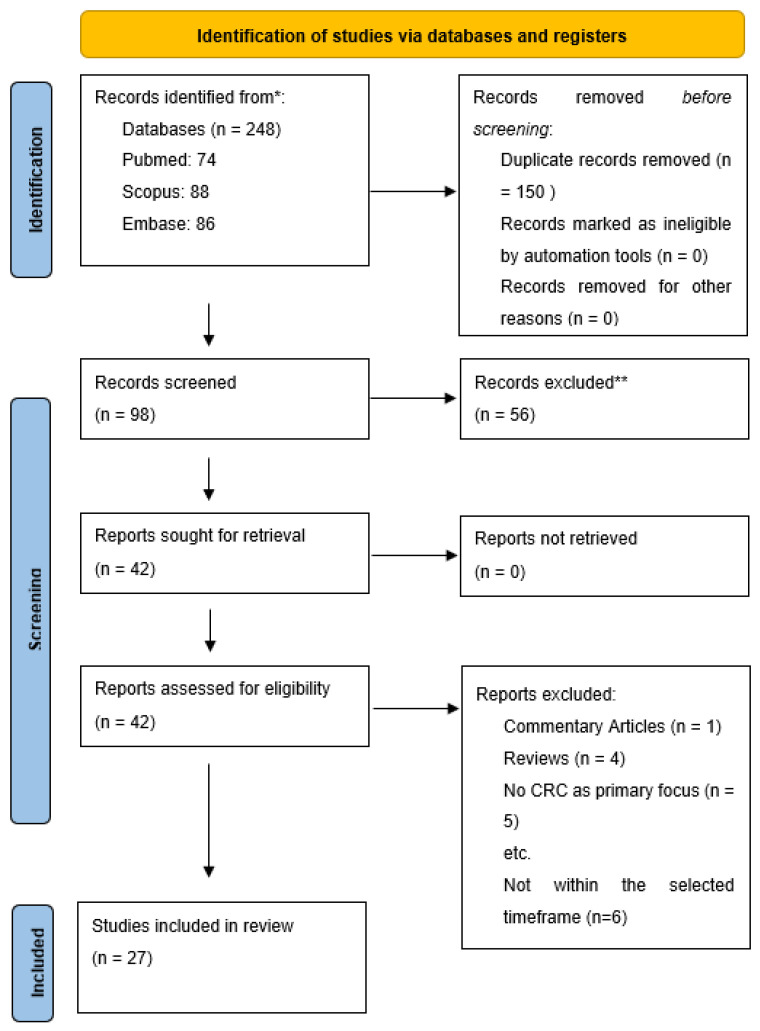
PRISMA Diagram. * Consider, if feasible to do so, reporting the number of “records identified from” each database or register searched (rather than the total number across all databases/registers). ** If automation tools were used, indicate how many records were excluded by a human and how many were excluded by automation tools.

**Figure 2 ijms-24-03472-f002:**
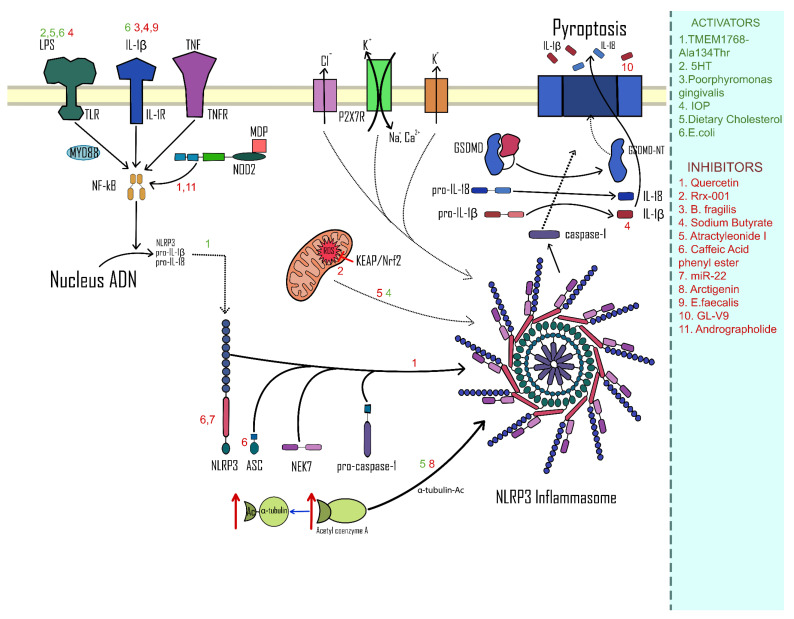
Pathways of activation/inhibition of the NLRP3 complex in CA-CRC.

## Data Availability

All of the referenced articles in this review are indexed to online datbases which are available for accessing.

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
