# Peer review of "The Multifaceted Role and Regulation of Nlrp3 Inflammasome in Colitis-Associated Colo-Rectal Cancer: A Systematic Review"

_ijms, 2023, doi:10.3390/ijms24043472_

Round 1
Reviewer 1 Report (New Reviewer)
For this article, I have only a few comments:
- I think it is better to remove table 1 from the main manuscript, and add it as supplement
- figure 1 should be changed, as it lacks quality and resolution
- more emphasis should be put in conclusions (especially) and/or discussion on future, clinical implications and theories that are important in NLRP3 inflammasome area of expertise
Author Response
Greetings, and we sincerely thank you for the invaluable feedback, which helps us improve our research. We will comment each suggestion point-by-point in the following paragraphs:
- I think it is better to remove table 1 from the main manuscript, and add it as supplement
We added that at the end, if available, we can upload it separately as well. The main reason is that we couldn't embed it in the body of the main manuscript because it was too large. In addition, the table presents with links to every included article, so it's important to maintain its' visibility. Thank you for the suggestion.
- figure 1 should be changed, as it lacks quality and resolution
We added a higher-resolution figure at the end. If required we can add that separately as well. We noticed that in the article, it dind't maintain it's pixel quality.
- more emphasis should be put in conclusions (especially) and/or discussion on future, clinical implications and theories that are important in NLRP3 inflammasome area of expertise
Based on your suggestion as well as your co-reviewer's suggestion we reshaped a part of the paragraphs to contain more references as well as a linear flow. In addition, each referenced paragraph has a conclusion, which highlights the importance and novelty of the referenced article. Not only that, we decided to improve our conclusion section, to emphasize the prognostic role as well as potential targeted treatment role. We believe that NLRP3 may be the key in improving the prognosis in colo-rectal cancer and CA-CRC as well as in inflammatory bowel diseases. Thank you for your suggestion!
Reviewer 2 Report (New Reviewer)
The review by Zaharie et al requires a number of minor points to be addressed before it is suitable for publication.
The primary roles of inflammasomes consist of: a) removal of the pathogenic agent, b) maintaining tissue homeostasis, c) removal of tumoral cells via an adaptive immune response [6].
Ref 6 focuses on assembly and activation of inflammasomes, the role of inflammasomes in neurologic disorders and metabolic diseases, and therapies targeting inflammasomes. Additional references are needed for points "a", "b", and "c".
The statement "Mutations of the NLRP3 inflammasome complex play an important role in the development of dominantly inherited autoinflammatory diseases, and its deregulation has been linked to carcinogenesis" needs to be referenced. For example Ref 17 can be cited here and [https://doi.org/10.1186/s12943-018-0900-3], which is more current, can also be cited.
The statement "This review will summarize the role of NLRP3 inflammasome complex in the development of CA-CRC in contrast with the recent pathways of activation or downregulation of the inflammasome complex, in order to develop therapeutic strategies in IBD as well as in CA-CRC" is difficult to follow. Since the paragraph ends with "This can further ramp up the role of NLRP3 as a prognostic factor in the aformentioned pathologies as well as can identify various methods of treatment in the chronic inflammation of the intestinal tract. Our focus will not be on the mechanisms of action, as this was discussed in the previous reviews [7,12,13]", the phrase "in contrast with the recent pathways of activation or downregulation of the inflammasome complex" can be deleted. For example, something like:
"This review will summarize the role of the NLRP3 inflammasome complex in the development of CA-CRC and summarize studies that report on the downregulation of NLRP3 activity."
In Figure 1, the "Records identified from" box should have an " * ".
Ref 14 uses "dextran sulfate sodium" not "dextrane sulfate sodium".
The statement "Caspase-1 will act as a catalyst in the activation of the IL-1β and IL-18 proteins into cytokines" is not clear. Caspase-1 cleaves IL-1 family precursor proteins to generate biologically active mature IL-1β and IL-18. If needed [https://doi.org/10.1186/s41232-019-0101-5] can be cited here.
The statement "The non-canonical pathway is activated by intracellular DAMPs, one example of that being the reactive oxygen species (ROS) [14,17,18]" suggests that reactive oxygen species are DAMPs. This is not correct, reactive oxygen species are not DAMPs. While Ref 17 does state that reactive oxygen species are DAMPs, the reference cited by Ref 17 is Cassel et al, 2009, and while Cassel et al, 2009, state that ROS can be involved in activation of NLRP3 they do not state that reactive oxygen species are DAMPs.
[https://doi.org/10.1186/s12943-018-0900-3] gives a good description of ROS and NLRP34 activation.
The sentence "In addition, a number of studies have focused on NLRP3 as a prognostic marker [24-26]" is not correct. Ref 26 states " Furthermore, we demonstrate that increased tumorigenesis is mediated through the NLRC4 inflammasome, rather than through NLRP3." Therefore, this sentence should cite only references 24 and 25.
The sentence "By harvesting human CRC tissue microarray (TMA) in order to obtain human colon carcinoma cells, such HCT116 and RKO, which were used to create xenografts (on mice), NLRP3 upregulation was associated with the presence of CRC, in comparison with precancerous tissue in which the NLRP3 levels were lower [24]" is confusing. The tissue microarrays were not harvested to obtain human colon carcinoma cell lines. Tissue microarrays were used for histochemical analysis of 60 pairs of adenocarcinoma and paracancerous tissues (not precancerous tissue). In the study, frozen human tissue samples were used for Western Blot analysis. The HCT116 and RKO cell lines were obtained from the Type Culture Collection of the Chinese Academy of Sciences (Shanghai, China). In 49 pairs of TMA samples NLRP3 was upregulated in the colon adenocarcinoma tissue, and NLRP3 was downregulated in 11 samples. NLPR3 was also upregulated in the frozen human tissue samples and in the xenographs. This is shown in Ref 25, not Ref 24. Finally, for the general reader, a reference for TMA can be given. Therefore, the sentence should be changed to something like:
"Analysis of frozen human tissue samples and mouse xenograph models using the human colon carcinoma cell lines HCT116 and RKO demonstrated that NLRP3 was upregulated in human colon carcinoma tissue [25]. Histochemical analysis of 60 pairs of human colon adenocarcinoma and paracancerous microarray samples showed NLRP3 upregulation in 49 adenocarcinoma samples and downregulation in 11 adenocarcinoma samples [25, https://doi.org/10.4103/0256-4947.51806]. Notably, other inflammasomes, such as NLRC4, can also be associated with inflammation-induced tumorigenesis [26]."
The text states "... further data being required to provide a link between the intestinal inflammation and the development of CRC. Higher expressions of CRC may lead to a poorer prognosis of the disease [26,27]." First, there is a massive amount of literature demonstrating a link between inflammation and cancer, including intestinal inflammation and the development of CRC. Therefore, the general statement that further data is required to provide a link between the intestinal inflammation and the development of CRC should be deleted. Second, there is no such thing as higher expression of colorectal carcinoma. Possibly the authors mean higher expression of NLPR3 rather than higher expression of CRC. If that is correct, the sentence "Higher expressions of NLPR3 may lead to a poorer prognosis of the disease [26,27]." is redundant and should be deleted.
The statement "High stromal levels of transmembrane protein 176B, a cationic channel is associated with low overall survival in patients with CRC. TMEM176B inhibits NLRP3-mediated IL-1B production in dendritic cells, thus lowering the tumoral infiltration and activation of CD8+ T cells. In addition, a minor-expression of TMEM176B may lead to an increase of IL-18 which has a beneficial effect on the gut mucosa. A protective variant of the protein, Ala134Thr has led to a diminished expression of TMEM176B, without managing to identify a cell-specific effect, due to its observation in tissue, not in isolated cells." is confusing. TMEM176B inhibits NLRP3 activation, therefore, the statement that inhibition of NLRP3 activation is associated with low patient survival appears to directly contradict the data presented in the previous paragraph, which indicates that increased NLRP3 activation is associated with low patient survival. Therefore, this paragraph needs to be rewritten:
"High stromal levels of transmembrane protein 176B, a cationic channel, is associated with low overall survival in patients with CRC. Since TMEM176B inhibits NLRP3 activation, this stands in seeming contrast to the results of the studies presented above. However, TMEM176B appears to primarily affect immune infiltrating cells and not the primary tumor....." At this point the authors can describe the effects of TMEM176B and the Ala134Thr variant. In addition, reference to inflammation and the tumor microenvironment can be made at this point.
The text states "In addition, by testing whether the blocking of the 5-HT receptors will downregulate the production of IL-1β, only 5-HTRA3 receptor significantly impaired the production." This is very important and should be referenced.
The text states "However, some polymorphisms of the NLRP3 gene which are associated with a lower expression of the protein can lead to dysfunctional activity of the inflammasome during the early disease activity which can lead to a weak inflammatory response [36,37]." The authors need to explain that a weak inflammatory response may impair the efficiency of bacterial clearance in the intestinal mucosa, resulting in sustained inflammation. Also, Ref 36 does not discuss NLRP3 polymorphisms and should not be cited when referring to NLRP3 polymorphisms.
The statement " Studies have shown a decrease of A. municiphila in patients with CRC, and supplementing with it suppressed colonic tumorigenesis in APC induced mice." needs to be referenced.
The statement "Thus, an NLRP3 deficiency in macrophages can attenuate the tumor suppressive effect of A. municiphila" can be linked back to the discussion regarding increased NLRP3 activation in immune cells being beneficial.
The paragraph beginning "Another study highlights that activation of the NLRP3 inflammasome complex via...." should begin with "In contrast to the studies presented above, Li et al, 2021, report that activation of the NLRP3 inflammasome complex via...."
The statement "Therefore, the impact of NLRP3 in CA-CRC remains controversial, reminding that some studies are also supporting the hypothesis that inflammasomes are having a protective effect in colitis [43]" should be replaced with something like "Other studies have also reported anti-tumorigenic functions of NLRP3 [43]. Thus, the role of NLRP3 in CA-CRC is not fully understood and NLRP3-mediated anti- and pro-tumorigenic mechanisms remain a critical area of research [43]." Note that the word "controversial" contradicts the statement in the Introduction "This review will summarize the role of NLRP3 inflammasome complex in the development of CA-CRC".
To make section 3.2 easier to follow, the authors should discuss NLPR3 pro-tumorigenic studies first and then NLPR3 anti-tumorigenic studies. Therefore, the paragraph about TMEM176B should come after the paragraph about cholesterol, then the paragraph about A. municiphila, and the section should end with the paragraph about the anti-tumorigenic role of NLPR3.
"E. faecalis can attenuate this response" needs to be referenced.
The text states "The main limitations of this study, however, were using a single strand of cancer cells (HCT116), and the levels of IL-1 β were analyzed [64]." Why is analyzing the levels of IL-1 β a limitation of the study?
The text states "Some authors suggest that inflammasome complex stimulation through inflammation and ulteriorly cancer". I don't know what is meant by "ulteriorly cancer".
Author Response
Greetings and first of all we would like to thank you for the invaluable feedback for our article. This is a well written and thorough review which helped us not only find some flaws in the design of our article but also managed to significantly improve our paper. We really appreciate for taking your time into writing such an exhaustive review.
Before we provide our point-by-point responses we would like to mention that based on your suggestions some of the reference numbers have changed, due to some additions of more references as well as changing the order of paragraphs. However, we kept the original comments for a better flow-of-discussion.
Point-by-point responses:
1. The primary roles of inflammasomes consist of: a) removal of the pathogenic agent, b) maintaining tissue homeostasis, c) removal of tumoral cells via an adaptive immune response [6].
Ref 6 focuses on assembly and activation of inflammasomes, the role of inflammasomes in neurologic disorders and metabolic diseases, and therapies targeting inflammasomes. Additional references are needed for points "a", "b", and "c".
We added three more references, focusing on the primary roles on the inflammasomes in order to support this statement. Thank you for the suggestion.
2. The statement "Mutations of the NLRP3 inflammasome complex play an important role in the development of dominantly inherited autoinflammatory diseases, and its deregulation has been linked to carcinogenesis" needs to be referenced. For example Ref 17 can be cited here and [https://doi.org/10.1186/s12943-018-0900-3], which is more current, can also be cited.
Reference 17 was switched for this paragraph as well as adding the reference you suggested. Thank you!
3.The statement "This review will summarize the role of NLRP3 inflammasome complex in the development of CA-CRC in contrast with the recent pathways of activation or downregulation of the inflammasome complex, in order to develop therapeutic strategies in IBD as well as in CA-CRC" is difficult to follow. Since the paragraph ends with "This can further ramp up the role of NLRP3 as a prognostic factor in the aformentioned pathologies as well as can identify various methods of treatment in the chronic inflammation of the intestinal tract. Our focus will not be on the mechanisms of action, as this was discussed in the previous reviews [7,12,13]", the phrase "in contrast with the recent pathways of activation or downregulation of the inflammasome complex" can be deleted. For example, something like: "This review will summarize the role of the NLRP3 inflammasome complex in the development of CA-CRC and summarize studies that report on the downregulation of NLRP3 activity."
We had issues writing that paragraph at first, however your suggestion fits perfectly, therefore we thank you for helping us improving this paragraph. We corrected that accordingly.
4. In Figure 1, the "Records identified from" box should have an " * ".
Added. Thank you.
5. Ref 14 uses "dextran sulfate sodium" not "dextrane sulfate sodium".
Corrected accordingly, most likely a typo. Thank you very much.
6.The statement "Caspase-1 will act as a catalyst in the activation of the IL-1β and IL-18 proteins into cytokines" is not clear. Caspase-1 cleaves IL-1 family precursor proteins to generate biologically active mature IL-1β and IL-18. If needed [https://doi.org/10.1186/s41232-019-0101-5] can be cited here.
Added both the reference and rephrased it for more clarity. Also added the reference, as well. Thank you for the suggestion.
7. The statement "The non-canonical pathway is activated by intracellular DAMPs, one example of that being the reactive oxygen species (ROS) [14,17,18]" suggests that reactive oxygen species are DAMPs. This is not correct, reactive oxygen species are not DAMPs. While Ref 17 does state that reactive oxygen species are DAMPs, the reference cited by Ref 17 is Cassel et al, 2009, and while Cassel et al, 2009, state that ROS can be involved in activation of NLRP3 they do not state that reactive oxygen species are DAMPs.
[https://doi.org/10.1186/s12943-018-0900-3] gives a good description of ROS and NLRP34 activation.
I believe the ROS part was added incorrectly, mostly we wanted to reference the fact that the non-canonical pathway is also caspase-11 dependent. We changed the references as well as removed the ROS part for this paragraph, in order for the readers to easily find and understand the non-canonical pathway. Thank you for pointing that!
8. The sentence "In addition, a number of studies have focused on NLRP3 as a prognostic marker [24-26]" is not correct. Ref 26 states " Furthermore, we demonstrate that increased tumorigenesis is mediated through the NLRC4 inflammasome, rather than through NLRP3." Therefore, this sentence should cite only references 24 and 25.
Fixed that as well. Thank you. I believe reference no 26 was supposed to be cited in another part, however we adjusted that part accordingly.
9.The sentence "By harvesting human CRC tissue microarray (TMA) in order to obtain human colon carcinoma cells, such HCT116 and RKO, which were used to create xenografts (on mice), NLRP3 upregulation was associated with the presence of CRC, in comparison with precancerous tissue in which the NLRP3 levels were lower [24]" is confusing. The tissue microarrays were not harvested to obtain human colon carcinoma cell lines. Tissue microarrays were used for histochemical analysis of 60 pairs of adenocarcinoma and paracancerous tissues (not precancerous tissue). In the study, frozen human tissue samples were used for Western Blot analysis. The HCT116 and RKO cell lines were obtained from the Type Culture Collection of the Chinese Academy of Sciences (Shanghai, China). In 49 pairs of TMA samples NLRP3 was upregulated in the colon adenocarcinoma tissue, and NLRP3 was downregulated in 11 samples. NLPR3 was also upregulated in the frozen human tissue samples and in the xenographs. This is shown in Ref 25, not Ref 24. Finally, for the general reader, a reference for TMA can be given. Therefore, the sentence should be changed to something like:"Analysis of frozen human tissue samples and mouse xenograph models using the human colon carcinoma cell lines HCT116 and RKO demonstrated that NLRP3 was upregulated in human colon carcinoma tissue [25]. Histochemical analysis of 60 pairs of human colon adenocarcinoma and paracancerous microarray samples showed NLRP3 upregulation in 49 adenocarcinoma samples and downregulation in 11 adenocarcinoma samples [25, https://doi.org/10.4103/0256-4947.51806]. Notably, other inflammasomes, such as NLRC4, can also be associated with inflammation-induced tumorigenesis [26]."
I believe that went as a translation typo. With this paragraph we wanted to highlight the upregulation of NLRP3 in carcinoma tissue. We changed that according to your suggestion and we thank you for that.
10. The text states "... further data being required to provide a link between the intestinal inflammation and the development of CRC. Higher expressions of CRC may lead to a poorer prognosis of the disease [26,27]." First, there is a massive amount of literature demonstrating a link between inflammation and cancer, including intestinal inflammation and the development of CRC. Therefore, the general statement that further data is required to provide a link between the intestinal inflammation and the development of CRC should be deleted. Second, there is no such thing as higher expression of colorectal carcinoma. Possibly the authors mean higher expression of NLPR3 rather than higher expression of CRC. If that is correct, the sentence "Higher expressions of NLPR3 may lead to a poorer prognosis of the disease [26,27]." is redundant and should be deleted.
I believe this was a typo due to the language barrier, because that’s exactly what we meant to say, regarding the link between the inflammation and cancer, especially in an inflammatory bowel disease. We thank you for pointing that out. Also, we repeated ourselves regarding the NLRP3 expression, therefore we adjusted the phrase accordingly.
11.
The statement "High stromal levels of transmembrane protein 176B, a cationic channel is associated with low overall survival in patients with CRC. TMEM176B inhibits NLRP3-mediated IL-1B production in dendritic cells, thus lowering the tumoral infiltration and activation of CD8+ T cells. In addition, a minor-expression of TMEM176B may lead to an increase of IL-18 which has a beneficial effect on the gut mucosa. A protective variant of the protein, Ala134Thr has led to a diminished expression of TMEM176B, without managing to identify a cell-specific effect, due to its observation in tissue, not in isolated cells." is confusing. TMEM176B inhibits NLRP3 activation, therefore, the statement that inhibition of NLRP3 activation is associated with low patient survival appears to directly contradict the data presented in the previous paragraph, which indicates that increased NLRP3 activation is associated with low patient survival. Therefore, this paragraph needs to be rewritten:
"High stromal levels of transmembrane protein 176B, a cationic channel, is associated with low overall survival in patients with CRC. Since TMEM176B inhibits NLRP3 activation, this stands in seeming contrast to the results of the studies presented above. However, TMEM176B appears to primarily affect immune infiltrating cells and not the primary tumor....." At this point the authors can describe the effects of TMEM176B and the Ala134Thr variant. In addition, reference to inflammation and the tumor microenvironment can be made at this point.
We’ve revamped the paragraph per your suggestion as well as adding more references and enhancing the paragraph by highlighting the effects of TMEM176B, as well as referencing the impact of tumor microenvironment.
12. The text states "In addition, by testing whether the blocking of the 5-HT receptors will downregulate the production of IL-1β, only 5-HTRA3 receptor significantly impaired the production." This is very important and should be referenced.
It was referenced, for some reason the number was not added. It has similar references with the previous paragraph. Thank you for reporting that!
13. The text states "However, some polymorphisms of the NLRP3 gene which are associated with a lower expression of the protein can lead to dysfunctional activity of the inflammasome during the early disease activity which can lead to a weak inflammatory response [36,37]." The authors need to explain that a weak inflammatory response may impair the efficiency of bacterial clearance in the intestinal mucosa, resulting in sustained inflammation. Also, Ref 36 does not discuss NLRP3 polymorphisms and should not be cited when referring to NLRP3 polymorphisms.
We removed the incorrect reference, as well as adding the discussion regarding the weak inflammatory response leading to sustained inflammation due to the inefficiency of bacterial clearence, with references.
14. The statement " Studies have shown a decrease of A. municiphila in patients with CRC, and supplementing with it suppressed colonic tumorigenesis in APC induced mice." needs to be referenced.
Added the references. This was due to the fact that we considered the entire paragraph as a reference to the article.
15. The statement "Thus, an NLRP3 deficiency in macrophages can attenuate the tumor suppressive effect of A. municiphila" can be linked back to the discussion regarding increased NLRP3 activation in immune cells being beneficial.
We tracked back the discussion as well as highlighting key references. Thank you for the suggestion.
16. The paragraph beginning "Another study highlights that activation of the NLRP3 inflammasome complex via...." should begin with "In contrast to the studies presented above, Li et al, 2021, report that activation of the NLRP3 inflammasome complex via...."
Corrected. Thank you for the suggestion
17. The statement "Therefore, the impact of NLRP3 in CA-CRC remains controversial, reminding that some studies are also supporting the hypothesis that inflammasomes are having a protective effect in colitis [43]" should be replaced with something like "Other studies have also reported anti-tumorigenic functions of NLRP3 [43]. Thus, the role of NLRP3 in CA-CRC is not fully understood and NLRP3-mediated anti- and pro-tumorigenic mechanisms remain a critical area of research [43]." Note that the word "controversial" contradicts the statement in the Introduction "This review will summarize the role of NLRP3 inflammasome complex in the development of CA-CRC".
We thank you for the heads-up. We used your suggestion in the paragraph
18. To make section 3.2 easier to follow, the authors should discuss NLPR3 pro-tumorigenic studies first and then NLPR3 anti-tumorigenic studies. Therefore, the paragraph about TMEM176B should come after the paragraph about cholesterol, then the paragraph about A. municiphila, and the section should end with the paragraph about the anti-tumorigenic role of NLPR3.
We changed the order of the paragraphs according to your suggestion. We initially wanted to group up the factors that upregulate NLRP3 based on their primary type (bacteria, synthetic, etc).
19. "E. faecalis can attenuate this response" needs to be referenced.
Again, it was probably due to the fact that the entire paragraph was referenced. We fixed that and we thank you.
20. The text states "The main limitations of this study, however, were using a single strand of cancer cells (HCT116), and the levels of IL-1 β were analyzed [64]." Why is analyzing the levels of IL-1 β a limitation of the study?The idea was incomplete. It was supposed to say “The main limitations of this study, were using a single strand of cancer cells (HCT116), and the levels of IL-1 β were analyzed using ELISA assays, however the effects of the stimulation of cells with IL-1β were not compared.[64].” We fixed that and added in the manuscript.
21. The text states "Some authors suggest that inflammasome complex stimulation through inflammation and ulteriorly cancer". I don't know what is meant by "ulteriorly cancer".
We meant to highlight the transition phase from inflammation to cancer, both of the phases stimulating the inflammasome complex, thus triggering the NK cell maturation. Per your suggestion, we can delete that part.
We again thank you for taking your time into reading our response as well as your invaluable suggestions which we implemented.
This manuscript is a resubmission of an earlier submission. The following is a list of the peer review reports and author responses from that submission.
Round 1
Reviewer 1 Report
I have reviewed the manuscript titled “The Multifaceted Role And Regulation of Nlrp3 Inflammasome 2 in Colitis-Associated Colo-Rectal Cancer: A Systematic Review”. The stated objective of the review is to summarize the role of NLRP3 inflammasome complex in the develelopment of CA-CRC. The search strategy is summarized and main studies reported.
Major comments:
1. Reporting of findings should be clarified and made more concise. Example is line 159 page 5. NLRP3 upregulation was discovered, especially when compared with precancerous tissue. This sentence should be reworded to clarify comparison made.
2. Line 166 page 5. It is not clear from the sentence if overexpression of NLRP3 is a prognostic marker for CRC or for the progression to CRC from IBD. Please clarify.
3. The role of gut microbiome on serotonin should be included.
Minor comments:
Several typos/grammatical issues: line 71, page 2, opinions is misspelled.
Author Response
We thank you for your thorough revision, as well as for your feedback, as it helped us find the flaws in our review. We will address each point individiually, step by step.
1. Reporting of findings should be clarified and made more concise. Example is line 159 page 5. NLRP3 upregulation was discovered, especially when compared with precancerous tissue. This sentence should be reworded to clarify comparison made.
I believe this issue stems from the fact that English is not our native language. This can also be associated with the minor issues. I rephrased the paragraph in order to be more clear and concise. What we meant with that phrase was the fact that there is a significant overexpression of NLRP3 in cancerous tissue compared to precancerous tissue, this overexpression being highlighted in some of the studies mentioned in the review. We have rephrased it accordingly.
2. Line 166 page 5. It is not clear from the sentence if overexpression of NLRP3 is a prognostic marker for CRC or for the progression to CRC from IBD. Please clarify.
A certain number of studies have shown that overexpression of NLRP3 is an independent prognostic marker in the development of colo-rectal cancer, as higher expressions of NLRP3 may lead to a poorer prognosis. There is still work to be done whether the overexpression of NLRP3 is a consequence mainly of the development of CRC or can be fully attributed to the inflammation in IBD. We have rephrased this accordingly. Thank you for highlighting this.
3. The role of gut microbiome on serotonin should be included.
One of the articles under the review focuses exactly on that aspect. Although it's not the main focus of the topic, we agree that some explanations in this aspect are required, therefore we added them.
We sincerely thank you for the positive feedback as well as for the suggestions. If there are any other things required, please let us know.
Reviewer 2 Report
Zaharie and coauthors present a literature review of the role of the NLRP3 inflammasome and it's role in the genesis of colorectal cancer. The design of their study is well thought out and the presentation of the data is fairly clear, although the manuscript could benefit from proofreading by a native English speaker to clear up some grammar/syntax issues.
However, the manuscript suffers in two ways, one minor and one fatal. To begin, the figures appear to be copied and pasted directly from Microsoft Powerpoint and Excel. Figure 1 includes the editing marks present onscreen for spelling and grammar. Table 1 contains line breaks in the cells that should be addressed by manually removal instead of just copy-pasting from another source i.e., online manuscript titles. Additionally, there are issues with bacterial naming - as an example the authors list names in the form of "E. Coli" when it should be "E. coli" or other organisms. These are minor, easily fixed issues.
The second, fatal problem is in regards to the number of source manuscripts used to conduct this review. For the manuscripts the authors reviewed, they present a clear and well-reasoned analysis. However, their search for relevant manuscripts was woefully inadequate. For a field ~20 years old, only finding 18 manuscripts to draw from shocked me. In about ten minutes of searching on Google scholar for "NLRP3 colorectal cancer, I found another 18 manuscripts the authors did not identify that are relevant to this work:
1. Du et al 2016. Dietary cholesterol promotes AOM-induced colorectal cancer through activating the NLRP3 inflammasome
2. Wang et al 2020. The association of aberrant expression of NLRP3 and p-S6K1 in colorectal cancer
3. Chung et al 2019. Pretreatment with a Heat-Killed Probiotic Modulates the NLRP3 Inflammasome and Attenuates Colitis-Associated Colorectal Cancer in Mice
4. Li et al 2021. Overproduction of Gastrointestinal 5-HT Promotes Colitis-Associated Colorectal Cancer Progression via Enhancing NLRP3 Inflammasome Activation
5. Fan et al 2021. A. Muciniphila Suppresses Colorectal Tumorigenesis by Inducing TLR2/NLRP3-Mediated M1-Like TAMs
6. Zaki et al 2010. IL-18 Production Downstream of the Nlrp3 Inflammasome Confers Protection against Colorectal Tumor Formation
7. Wang et al 2016. Inflammasome-independent NLRP3 is required for epithelial-mesenchymal transition in colon cancer cells
8. Cambui et al 2022. The Ala134Thr variant in TMEM176B exerts a beneficial role in colorectal cancer prognosis by increasing NLRP3 inflammasome activation
9. Liu et al 2021. Sodium butyrate inhibits colitis-associated colorectal cancer through preventing the gut microbiota dysbiosis and reducing the expression of NLRP3 and IL-1β
10. Perera et al 2017. NLRP3 Inhibitors as Potential Therapeutic Agents for Treatment of Inflammatory Bowel Disease
11. Cong et al 2020. miR-22 Suppresses Tumor Invasion and Metastasis in Colorectal Cancer by Targeting NLRP3
12. Lee et al 2022. Inhibition of NLRP3 by Fermented Quercetin Decreases Resistin-Induced Chemoresistance to 5-Fluorouracil in Human Colorectal Cancer Cells
13. Zhang et al 2022. NLRP3 Inflammasome Activation in MΦs-CRC Crosstalk Promotes Colorectal Cancer Metastasis
14. Reid et al 2022. The small molecule NLRP3 inhibitor RRx-001 potentiates regorafenib activity and attenuates regorafenib-induced toxicity in mice bearing human colorectal cancer xenografts
15. Dagenais and Saleh 2015. Linking cancer-induced Nlrp3 inflammasome activation to efficient NK cell-mediated immunosurveillance
16. Yao et al 2017. Remodelling of the gut microbiota by hyperactive NLRP3 induces regulatory T cells to maintain homeostasis
17. Allen et al 2010. The NLRP3 inflammasome functions as a negative regulator of tumorigenesis during colitis-associated cancer
18. Ungerback et al 2012. Genetic variation and alterations of genes involved in NFκB/TNFAIP3- and NLRP3-inflammasome signaling affect susceptibility and outcome of colorectal cancer
It is not clear to me why the authors did not include these works (or indeed, any others that I did not find because I stopped looking), but this work is far from a systematic review of the available literature.
Author Response
We thank you for your well written and thorough review. It has proven to be invaluable to our work and spotted a certain number of mistakes which we're hopeful we can fix in a timely manner. We will approach every remark in a step-by-step fashion.
To begin, the figures appear to be copied and pasted directly from Microsoft Powerpoint and Excel. Figure 1 includes the editing marks present onscreen for spelling and grammar. Table 1 contains line breaks in the cells that should be addressed by manually removal instead of just copy-pasting from another source
We attempted to fix that earlier, however we were not familiar with the current format of the manuscript. This will get fixed, and we thank you for highlighting that.
Additionally, there are issues with bacterial naming - as an example the authors list names in the form of "E. Coli" when it should be "E. coli" or other organisms. These are minor, easily fixed issues.
I believe those were typos written in the editing program (the auto-capitalization after a dot). This was easily fixed by re-reading the entire document and finding the mistakes.
he second, fatal problem is in regards to the number of source manuscripts used to conduct this review. For the manuscripts the authors reviewed, they present a clear and well-reasoned analysis. However, their search for relevant manuscripts was woefully inadequate. For a field ~20 years old, only finding 18 manuscripts to draw from shocked me. In about ten minutes of searching on Google scholar for "NLRP3 colorectal cancer, I found another 18 manuscripts the authors did not identify that are relevant to this work:
We've looked into the list you provided, and revised our search and exclusion criteria, to see whether we made some mistakes or not. It was perhaps our fault that some criteria were not made clear.
First, these following articles did not fit the timeframe we've initially established, since the search criteria was over the last 10 years.
- Allen et al 2010. The NLRP3 inflammasome functions as a negative regulator of tumorigenesis during colitis-associated cancer
- Zaki et al 2010. IL-18 Production Downstream of the Nlrp3 Inflammasome Confers Protection against Colorectal Tumor Formation
- Ungerback et al 2012. Genetic variation and alterations of genes involved in NFκB/TNFAIP3- and NLRP3-inflammasome signaling affect susceptibility and outcome of colorectal cancer
Second, a part of the articles you mentioned were already in the review table:
- Li et al 2021. Overproduction of Gastrointestinal 5-HT Promotes Colitis-Associated Colorectal Cancer Progression via Enhancing NLRP3 Inflammasome Activation
- Wang et al 2016. Inflammasome-independent NLRP3 is required for epithelial-mesenchymal transition in colon cancer cells
-
Liu et al 2021. Sodium butyrate inhibits colitis-associated colorectal cancer through preventing the gut microbiota dysbiosis and reducing the expression of NLRP3 and IL-1β
- Cong et al 2020. miR-22 Suppresses Tumor Invasion and Metastasis in Colorectal Cancer by Targeting NLRP3
These two articles did not fit the selection criteria for our articles (one of them being a review, and the other was not implemented based on the abstract)
- Yao et al 2017. Remodelling of the gut microbiota by hyperactive NLRP3 induces regulatory T cells to maintain homeostasis
- Perera et al 2017. NLRP3 Inhibitors as Potential Therapeutic Agents for Treatment of Inflammatory Bowel Disease
In addition, two articles were referenced and even discussed but it was not in the table. We apologize for that mistake and we thank you for highlighting it.
- Dagenais and Saleh 2015. Linking cancer-induced Nlrp3 inflammasome activation to efficient NK cell-mediated immunosurveillance
- Wang et al 2020. The association of aberrant expression of NLRP3 and p-S6K1 in colorectal cancer
Two issues remain regarding the 2022 referenced articles. The following articles didn't fit in the study because when we estabilished the timeframe for the articles, they weren't indexed into the databases. Notwithstanding, we intend to implement them into our review:
- Lee et al 2022. Inhibition of NLRP3 by Fermented Quercetin Decreases Resistin-Induced Chemoresistance to 5-Fluorouracil in Human Colorectal Cancer Cells
-
Zhang et al 2022. NLRP3 Inflammasome Activation in MΦs-CRC Crosstalk Promotes Colorectal Cancer Metastasis – July 2022
- Cambui et al 2022. The Ala134Thr variant in TMEM176B exerts a beneficial role in colorectal cancer prognosis by increasing NLRP3 inflammasome activation
- Zhang et al 2022. NLRP3 Inflammasome Activation in MΦs-CRC Crosstalk Promotes Colorectal Cancer Metastasis – July 2022
- Reid et al 2022. The small molecule NLRP3 inhibitor RRx-001 potentiates regorafenib activity and attenuates regorafenib-induced toxicity in mice bearing human colorectal cancer xenografts
The remaining four articles that were mentioned, were revised, and upon further reading we decided to implement them into the review. We're not sure how these didn't get through the selection criteria (most likely human error), but we thank you for noticing that. We've added them into the current review.
- Du et al 2016. Dietary cholesterol promotes AOM-induced colorectal cancer through activating the NLRP3 inflammasome
-
Chung et al 2019. Pretreatment with a Heat-Killed Probiotic Modulates the NLRP3 Inflammasome and Attenuates Colitis-Associated Colorectal Cancer in Mice
- Fan et al 2021. A. Muciniphila Suppresses Colorectal Tumorigenesis by Inducing TLR2/NLRP3-Mediated M1-Like TAMs
After we revised the search criteria and the abstracts we found one more article which would be suitable for our review as well as within the timeframe we selected. Therefore, these articles as well as the articles from the 2022 that were not referenced initially will be implemented into the current review. Therefore, our final study reviews 28 articles over a span of 10-years.
We thank you for highlighting these aspects, and if any other suggestions are required, your feedback will be highly appreciated.
Round 2
Reviewer 2 Report
While I appreciate the authors' responses to my criticisms of this manuscript, I nevertheless view this work as not meriting the status of comprehensive review, for two reasons:
First, it appears as though the authors have very narrowly defined what their topic is, in order to fit in the space between existing reviews (which they do reference in their manuscript).
Second, and most importantly, if a given topic only has eighteen published papers in ten years it is not an area of research that warrants a comprehensive review. A researcher can read all of the papers on the subject in a long weekend. Even if the references that were indexed after the authors submitted their original manuscript were to be included, the total of ~20 papers would not rise to the level of an area of research worth a review.
Author Response
We thank you again for the quick and thorough explanations regarding the manuscript and we're sorry that our submission is not suitable according to your requirements. We're going to reply to every comment in a similar fashion as our previous reply, so that it will be much easier to follow.
First, it appears as though the authors have very narrowly defined what their topic is, in order to fit in the space between existing reviews (which they do reference in their manuscript).
The main purpose of the review was choosing the pathways for activation and inhibition of the NLRP3 inflammasome in colitis-associated cancer as well as highlighting its role in the development or inhibition of liver metastases. We intended to solely focus on the articles that are focusing on the involvement of NLRP3 in this pathology. We tried to avoid overlaying with other reviews, therefore the "narrowing" of this aspect.
Second, and most importantly, if a given topic only has eighteen published papers in ten years it is not an area of research that warrants a comprehensive review. A researcher can read all of the papers on the subject in a long weekend. Even if the references that were indexed after the authors submitted their original manuscript were to be included, the total of ~20 papers would not rise to the level of an area of research worth a review.
As far as we know, there is no minimum number of articles that are required to perform a systematic review. The fact that there are 28 articles in this domain can also highlight the need of further research in that direction, as stated in our conclusions. We were not aware of such criteria being required for submitting a review.